# Protocol for PD SENSORS: Parkinson's Disease Symptom Evaluation in a Naturalistic Setting producing Outcome measuRes using SPHERE technology. An observational feasibility study of multi-modal multi-sensor technology to measure symptoms and activities of daily living in Parkinson's disease

Catherine Morgan [1,2] Ian Craddock,[3] Emma L Tonkin,[3] Kirsi M Kinnunen,[4] Roisin McNaney,[3] Sam Whitehouse,[3] Majid Mirmehdi,[3] Farnoosh Heidarivincheh,[3] Ryan McConville,[3] Julia Carey,[3] Alison Horne,[5] Michal Rolinski [1,2] Lynn Rochester,[6,7] Walter Maetzler,[8] Helen Matthews,[9] Oliver Watson,[10] Rachel Eardley,[3] Alan L Whone[1,2]

For numbered affiliations see end of article.

**Correspondence to**
Dr Catherine Morgan;
catherine.morgan@bristol.ac.uk

## ABSTRACT

**Introduction** The impact of disease-modifying agents on disease progression in Parkinson's disease is largely assessed in clinical trials using clinical rating scales. These scales have drawbacks in terms of their ability to capture the fluctuating nature of symptoms while living in a naturalistic environment. The SPHERE (Sensor Platform for HEalthcare in a Residential Environment) project has designed a multi-sensor platform with multimodal devices designed to allow continuous, relatively inexpensive, unobtrusive sensing of motor, non-motor and activities of daily living metrics in a home or a home-like environment. The aim of this study is to evaluate how the SPHERE technology can measure aspects of Parkinson's disease.

**Methods and analysis** This is a small-scale feasibility and acceptability study during which 12 pairs of participants (comprising a person with Parkinson's and a healthy control participant) will stay and live freely for 5 days in a home-like environment embedded with SPHERE technology including environmental, appliance monitoring, wrist-worn accelerometry and camera sensors. These data will be collected alongside clinical rating scales, participant diary entries and expert clinician annotations of colour video images. Machine learning will be used to look for a signal to discriminate between Parkinson's disease and control, and between Parkinson's disease symptoms 'on' and 'off' medications. Additional outcome measures including bradykinesia, activity level, sleep parameters and some activities of daily living will be explored. Acceptability of the technology will be evaluated qualitatively using semi-structured interviews.

### Strengths and limitations of this study

► The location for testing (Sensor Platform for HEalthcare in a Residential Environment/SPHERE house) allows free living while providing some ground truth through colour video capture of participants' activities for clinician annotation.

► Due to the multisensor data collection platform within the SPHERE house, the performance of machine learning models which are based on single or multiple modalities could be assessed.

► The qualitative tools will add the important dimension of the person with Parkinson's perspective on this technology.

► The small number of participants means that results generated from this study will not be generalisable to the wider population of people with Parkinson's disease (PD).

► The feasibility nature of Parkinson's Disease Symptom Evaluation in a Naturalistic Setting producing Outcome measuRes using SPHERE technology means that cost effectiveness will not be explored at this stage.

**Ethics and dissemination** Ethical approval has been given to commence this study; the results will be disseminated as widely as appropriate.

## INTRODUCTION

Parkinson's disease (PD) is the second most common neurodegenerative disease in the

UK, affecting more than 127 000 people. It is a chronic, progressive, disabling disease which is characterised by a large variety of possible motor symptoms (including bradykinesia, rigidity and tremor) and non-motor symptoms (including sleep disturbance, cognitive impairment and pain).[1] It is estimated that by the time of diagnosis with PD most patients have already lost around 50% of their dopaminergic neurons,[2] therefore attempts to modify the disease should be aimed at preclinical or early-stage disease states.

The search for a treatment which could modify PD, a so-called disease-modifying therapy or 'DMT', has thus far proven fruitless with no DMT reaching market authorisation. This is despite multiple potential therapies having been tested in randomised, double-blind clinical trials.[3]

Clinical trials testing DMTs have often used a clinical rating scale to quantify symptoms and monitor response to treatment as a primary outcome measure.[3] The gold-standard rating scale is the Movement Disorder Society-sponsored revision of the Unified Parkinson's Disease Rating Scale (MDS-UPDRS).[4] This is a validated[5] tool which evaluates a comprehensive spectrum of symptoms and activities. However, it is also a subjective[6] and non-linear[7] tool which exposes the study participant to the risk of observer bias and the 'Hawthorne Effect'.[8] It is used episodically, giving a snapshot of a person with PD, which may not capture the day-to-day and hour-to-hour fluctuations of this complex disease. The testing location is often an artificial environment like a clinic or laboratory. This can increase costs[9] and impair the ability of studies to appraise important activities of daily living (ADLs) such as engagement in social activities,[10] rare events such as falls[11] and other metrics which affect well-being and quality of life such as sleep.[12]

Development of technologies for the measurement of various aspects of PD has been evolving over the past 15 years.[13] Particularly motor symptoms, including gait and bradykinesia, are amenable to measurement by technology[14] continuously and over long time periods. For example, the Parkinson's KinetiGraph wearable movement recording device (developed by Global Kinetics) can measure tremor, bradykinesia and dyskinesia[15] and the Microsoft Kinect motion sensor can classify PD stages relating to gait impairment.[16] However, these tools are still only partially integrated into clinical trials, with most continuing to use periodic clinical rating scale assessments in the non-ecological setting as primary or secondary endpoints.[17] There are excellent initiatives which are guiding this field forward, for example a 'roadmap' for implementation of digital outcome measures using technologies,[18] which are tackling the important questions surrounding regulatory approvals and collaboration with industry.

Lacunae in the literature about free-living technology-assisted outcomes in PD include the measurement of non-motor outcomes[14] such as constipation and ADLs. The frequent use of wearable devices which are attached to specific body parts means that the information gathered is restricted to the movement in that part of the body; this may be inadequate fully to understand the context in which a measurement is taken and may result in low correlations with measures of functional disability and/or quality of life.[19] In addition, it can be challenging to establish and operationalise the 'ground truth' for benchmarking novel measurements. Reviews of technology-assisted outcome measures in a free-living environment have highlighted that to date there is no fully validated system which evaluates clinical features of PD in a naturalistic/ecologically valid (ie, home or home-like) environment.[20]

## Sensor platform for healthcare in a residential environment

The long-standing SPHERE (Sensor Platform for HEalthcare in a Residential Environment) project from the University of Bristol has led to the design of a multisensor platform with multimodal devices designed to allow continuous, relatively inexpensive, unobtrusive sensing of motor, non-motor and ADL metrics in a home or a home-like environment.[21 22]

The SPHERE sensor platform is embedded into a test-bed location called the 'SPHERE house'. This is a two-bedroom, two-storey property with kitchen, living room, dining room, bathroom and toilet.

Within the SPHERE house there are a number of sensors which include: wearable devices with accelerometers (in-house developed); environmental/ambient sensors which measure temperature, humidity and light levels; RGB-D ('red green blue-depth', referring to the three colour planes that together can generate the colour of a pixel in an image, in addition to the recording of distances to surfaces using infrared light) camera sensors that generate silhouettes of participants and bounding boxes that, when linked with the SPHERE wearable, let the system know who the participant is; appliance sensors which are able to tell when and how frequently an appliance is switched on and off; PIR (Passive InfraRed) sensors which detect motion in each room.

The SPHERE sensor platform has undergone extensive testing with healthy volunteers, being deployed to 50 homes over a year from January 2017. High-quality qualitative work has been conducted looking at the design and acceptability of the SPHERE sensors and technology in a person's life.[23 24] In addition, the SPHERE platform is being used in three clinical projects which are at different stages of completion, looking at outcomes around cardiac valve replacement, hip and knee surgery,[25] and in dementia/mild cognitive impairment.

The multimodal multisensor design of the SPHERE system[22] enables a richer and more fine-grained picture of free-living symptoms and activities to be collected. For example, the combined use of the wearable devices with accelerometers, alongside the camera and environmental sensors, enables machine-learning (ML) algorithms to opportunistically and continuously fuse data to build a time series of ADLs for each person in the home.[26 27] This

offers an exciting opportunity to conduct free-living evaluation of people with PD.

## PD symptom evaluation in a naturalistic setting producing outcome measures using SPHERE technology

Given the above gaps in knowledge, and appreciating the SPHERE system's capability, it is believed that SPHERE technology could be exploited for filling some of the gaps in the current use of technology to measure outcomes in PD (eg, for non-motor symptoms and ADLs). SPHERE technology has the potential to complement and enhance the existing clinical rating scales used to evaluate PD. PD Symptom Evaluation in a Naturalistic Setting producing Outcome measuRes using SPHERE technology (PD SENSORS) is a feasibility and acceptability study using the SPHERE platform to develop novel outcome measures in PD. It is designed with the intent that it will inform a future larger scale and longer-term investigation, in which SPHERE technology will be deployed to multiple homes of people with PD (protocol V.1.3; August 2020). This is a truly collaborative study with not only strong interdisciplinary collaboration between computer scientists and movement disorders clinicians but also industry involvement. The university academic team are working with a UK-based imaging and digital biomarker analysis company, IXICO, that contributes clinical and data science expertise and is providing additional wearable devices from Axivity (AX3) and Activinsights (GENEActiv Original) and gait mat technology from ProtoKinetics (Zeno instrumented walkway), to further enrich the dataset.

## Aims
### Primary aims

To understand how the raw sensor data obtained from the SPHERE system can be processed to translate into meaningful data for the clinical scientist on metrics of:

► Total body movement.
► Bradykinesia.
► Sleep quantity and quality.
► Urinary dysfunction.
► Meal preparation/consumption.
► Cleaning tasks.
► Getting out of a chair.
► Activity level (ambulatory activity and sedentary behaviour).
► Room to room transfers and time spent in each room.

### Secondary aims

1. To evaluate the acceptability of the continuous use of multiple and varied SPHERE sensors over a 5-day period in a home-like setting for people with PD and to explore the participants' experience of such a study.
2. To explore the use of home sensing as a means to discriminate between participants with PD and healthy control participants.
3. To assess the criterion validity of the technology-assisted outcomes by mapping the sensor data outcomes to

the gold-standard clinical rating scale (MDS-UPDRS) and patient-reported outcome measure (Parkinson's Disease Questionnaire-39/PDQ-39) when participants with PD are in both 'on' and 'off' medication states.

## METHODS
### Design

A small-scale feasibility and acceptability study.

### Setting and participants

The setting will be the SPHERE house: a customised two-bedroom residential house in Bristol, UK, with multiple embedded sensors and data collection tools.

Twelve pairs of participants comprising a person with PD and a control participant (likely to be a spouse, other family member, friend or carer) will be recruited to stay in the SPHERE house for 5 consecutive days.

Participants with PD will be eligible to enter the study if they meet certain inclusion criteria, including:

► Diagnosis of idiopathic PD according to UK Brain Bank Criteria.[28]
► Age over 18.
► Modified Hoehn and Yahr (H&Y) Scale score of 3 or less in 'off' state (ie, when they withhold dopaminergic medications).

Inclusion criteria for control participants include:

► No history of PD, REM (rapid eye movement) sleep behaviour disorder, dementia or other neurodegenerative/significant musculoskeletal condition.

The study's exclusion criteria include:

► Montreal Cognitive Assessment Score <26.
► Current active depressive symptoms; Beck Depression Inventory II score >19.
► Use of walking aids while inside a house to aid mobility either 'on' or 'off' medications.

Exclusion criteria were largely chosen to reduce the potential impact on sensor data of other factors, such as mobility aids altering accelerometry data making it difficult to evaluate bradykinesia.

Participants will be recruited through movement disorders specialist or general neurology outpatient clinics in North Bristol National Health Service (NHS) Trust and through posters in the outpatient department of North Bristol NHS Trust, the Cure Parkinson's Trust (CPT), a local Patient and Public Involvement (PPI) group or by word-of-mouth.

The medical PD care for those study participants with the disease will continue unchanged.

The study will be conducted in accordance with the Declaration of Helsinki and Good Clinical Practice Guidelines. Written informed consent will be obtained from all study participants before any study-related procedures are performed.

Full approval from the NHS Wales Research Ethics Committee 6 was granted on 17th of December 2019, and Health Research Authority and Health and Care Research Wales approval confirmed on 14th of January 2020.

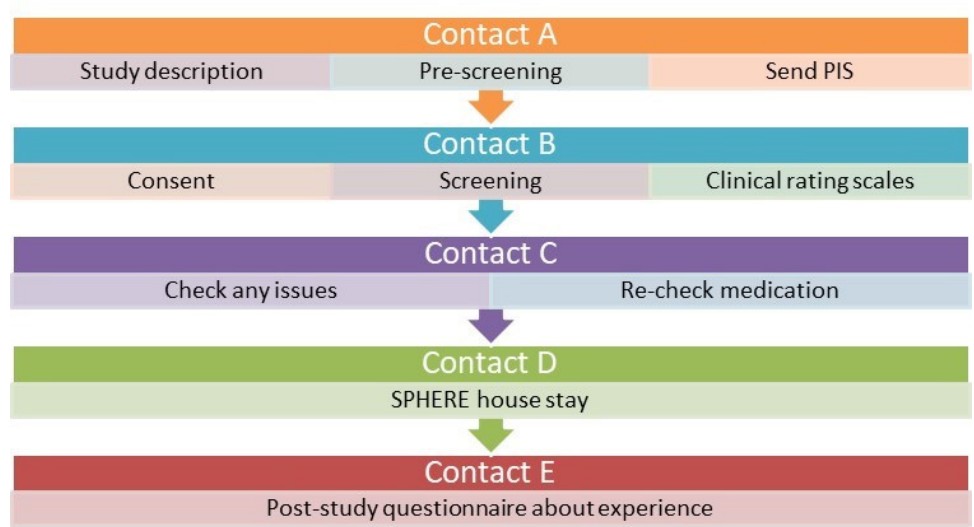

**Figure 1** Flow chart detailing outline of PD SENSORS study contacts. PD SENSORS, PD Symptom Evaluation in a Naturalistic Setting producing Outcome measuRes using SPHERE technology; PIS, Participant Information Sheet; SPHERE, Sensor Platform for HEalthcare in a Residential Environment.

Due to delays relating to the COVID-19 pandemic, active study data gathering is planned to commence in October 2020 and, if uninterrupted further, should be completed within 12 months.

### Sample size

This sample size of 24 was chosen as a balance between gathering enough data from the SPHERE sensors to create excellent quality algorithms (in consultation with our Machine Learning team) and the need to pragmatically take into account the nature of this exploratory study which is developing analytics to use in deployment to people's own homes in a future study. No conventional sample size calculation has been undertaken since we have no clear effect estimates to use in any such calculation; we view this more as a pilot and a proof of concept study. The comparator of the healthy volunteer control participant was chosen in order to make the study more sociable and therefore hopefully enjoyable to the person with PD, alongside also providing study data from people without PD so that conclusions could be drawn about how to identify and then measure aspects of PD.

### Overview of study contacts

PD SENSORS is split into five study 'contacts' (stages at which information is transmitted to or gathered from the study participants). A flow chart in figure 1 below illustrates the main aims of each study contact.

At each study contact, specific information is gathered from the participants by the research team. Table 1 details the tools to be used during the longer study contacts, contacts B and D, with the participants. The items in italics in this table apply only to the participant with PD and not to the control participant.

### Clinical rating scales

Several clinical rating scales will be completed as described at contact B. This is with the intent to gain a thorough understanding of the nature of the symptoms experienced by the participants with PD, and to be able to therefore make better sense of the sensor data produced from the SPHERE house stay. The Parkinson's Disease Sleep Scale-2 is a 15-point visual analogue scale that has been validated in PD[29] which allows patients with PD to self-rate and quantify the profile of nocturnal disturbances and sleep disruption being experienced. The Non-Motor Symptoms Scale is a 30-question scale which assesses severity and frequency with which a patient with PD experiences a variety of non-movement related symptoms, such as urinary dysfunction, mood/cognitive symptoms and gastrointestinal tract-associated symptoms.[30] The Timed Up and Go test[31] is a simple, quick and widely used clinical performance tool which assesses lower extremity function, mobility and falls risk. The REM sleep Behaviour Disorder Screening Questionnaire is a useful screening tool due to its relatively high sensitivity for REM sleep behaviour disorder,[32] which affects a significant proportion of patients with PD and may change the interpretation of the sensor data overnight.

During contact D, the research team will conduct the MDS-UPDRS[5] with both participants. This clinical rating scale has subscales evaluating motor experiences of daily living (part I), non-motor experiences of daily living (part II), motor examination (part III) and motor complications (part IV). The PDQ-39, also to be used in contact D, is a self-completed patient-reported outcome measure designed to address aspects of functioning and well-being for those affected by PD.[33]

**Table 1** Baseline and follow-up data from contacts B and D

| Time point | Outcomes measured |
|---|---|
| Contact B | Further demographic information including date of birth and *date of diagnosis of PD* |
| | Full screening questions according to inclusion and exclusion criteria |
| | *Medication regime and calculation of Levodopa equivalent daily dose* |
| | *H&Y score* |
| | *MoCA* |
| | *BDI II, AS, ESS* |
| | Timed Up and Go Test |
| | *NMSS* |
| | *PDSS-2* |
| | *RBD-SQ* |
| Contact D (the SPHERE house stay) | MDS-UPDRS (the motor subscale will be performed on multiple occasions, *once while 'off' medications*) |
| | PDQ-39 |
| | Symptom and activity diary |
| | Sleep diary |
| | Bladder diary |
| | *Medication-taking record* |
| | Sensor data from scripted activities |
| | Sensor data from free-living |
| | Sensor data from wearable devices provided through IXICO |
| | Annotations of colour video dataset |
| | Gait mat assessments |
| | Interview with questions relating to experience of living with SPHERE technology |

AS, Apathy Scale; BDI BDI II, Beck's Depression Inventory II; ESS, Epworth Sleepiness Scale; H&Y, Hoehn and Yahr; MDS-UPDRS, Movement Disorder Society-sponsored revision of the Unified Parkinson's Disease Rating Scale; MoCA, Montreal Cognitive Assessment; NMSS, Non-Motor Symptoms Scale for Parkinson's disease; PD, Parkinson's disease; PDQ-39, Parkinson's Disease Questionnaire-39; PDSS-2, Parkinson's Disease Sleep Scale-2; RBD-SQ, REM sleep Behaviour Disorder Screening Questionnaire; SPHERE, Sensor Platform for HEalthcare in a Residential Environment.

### Schedule of contact D: the stay in the SPHERE house

The pair of participants will stay in the house together for 5 days. The duration of stay was chosen in consultation with people with PD and their acquaintances who took part in prestudy focus groups. During the time they are in the house, the participants will be encouraged to continue living freely and undertake their regular ADLs as normally as possible.

Figure 2 shows a day-by-day plan for contact D for the pair of participants. The figure illustrates that the stay in the SPHERE house for each pair of participants will also involve:

► Researchers will visit the house on two occasions, day 1 and day 4, to conduct scripted activities and clinical rating scales with participants.
► During days 2 and 3, the participants will live freely with no researcher visits.

Important elements during the 5-day stay are:

► The participant with PD will withhold their dopaminergic medications for >12 hours so that they are in the practically defined 'off' medication state. During this time 'off' medications (on day 4), some assessments will be undertaken including the MDS-UPDRS III (motor subsection). These assessments will then be repeated after the participant with PD restarts their medications that day.
► Specific food preparation and cleaning/hygiene scripted activities, will be witnessed and annotated by a study team member during their planned visits to the house. The participant with PD will undertake these activities on day 1 and twice on day 4, first 'off' then 'on' medications, while the control participant will do the activities once each on day 1 and day 4. More detail about these activities is shown in the online supplemental material 1.
► Continuous sensor data will be captured throughout the 5-day stay from the following SPHERE sensors: cameras (that provide silhouettes and bounding boxes), appliance sensors, environment/ambient sensors, PIR sensors.
► Wearable devices: in addition to the SPHERE wearable device on each wrist recording continuous accelerometry, an extra IXICO-provided wearable will be worn on each wrist throughout the 5-day stay. During the clinical assessments and scripted activities on day 4, one additional wearable accelerometer will be placed around the trunk to locate in the lower back (lumbar) area and another one on a lower limb (shin).

### Gait mat assessments

During the 'off' and 'on' medication testing session on day 4, the participants will undertake clinical assessments including an evaluation of gait (normal pace, slow pace and fast pace) on the Zeno Walkway, a gait mat embedded with sensors that is placed in the entrance area of SPHERE house where these assessments are conducted. The item is administered according to the standard instructions and the gait mat protocol. All gait mat protocols for these assessments have been developed by ProtoKinetics, the Zeno Walkway manufacturers.

### Colour video capture

In communal areas of the SPHERE house (ie, not in bedrooms or bathrooms) cameras will be enabled to produce RGB data for the purposes of allowing more accurate and relevant evaluation of the participants' movements. Short periods of these data, collected both while participants are free living and during clinical

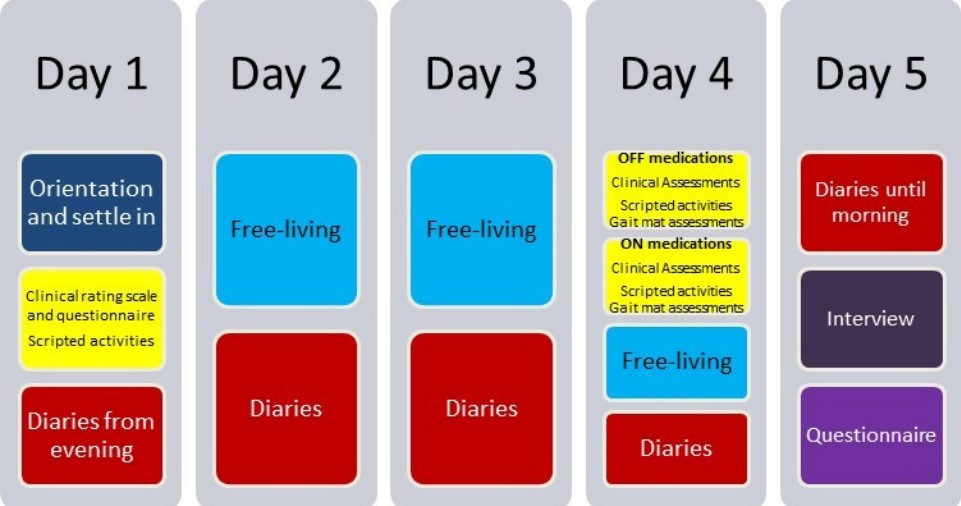

**Figure 2** A visualisation of the day-to-day participant activities during contact D, the SPHERE house stay. OFF medications= the participant with PD in practically defined 'off' medication state which entails withholding dopaminergic medications for a period of more than 12 hours; ON medications= the participant with PD having taken their dopaminergic medications. PD, Parkinson's disease; SPHERE, sensor platform for healthcare in a residential environment.

assessments, will be used to provide a labelled dataset for the analytic work-up of total body movement using annotations of these videos relating to the participants' activities, location and aspects of their PD symptomatology.

All work during the events and circumstances relating to SARS-CoV-2/COVID-19 will be subject to risk assessment and modification to ensure the study is safe for participants and researchers.

### Data analysis and management
#### Statistical data analysis approaches

The purpose of this preliminary and exploratory study is to explore the best way to quantify PD using the in-house technology, therefore the exact nature of the analytical approach(es) is not available at this stage. Study researchers and the study statistician will work collaboratively with the clinicians and SPHERE ML team to determine how the SPHERE data can be transformed into usable form. The research team will then aim to ascertain if this data can give us a signal to discriminate between participants with PD and control participants and whether it can give us a signal to discriminate between when a participant with PD is 'on' and 'off' medications. We will compare it to the data produced by the gold-standard clinical rating scales and patient-reported outcome measures. ML will play a key role in analysing the data.

Existing works have proposed methods based on ML for PD diagnosis and monitoring (please refer to[34] for a detailed review). Using various types of data, such as videos, Inertial Measurement Unit and smartphone or wearable accelerometer data, they model the presence and/or severity of PD via ML algorithms such as Artificial Neural Networks, Hidden Markov Models (HMM), Support Vector Machines (SVM) and Random Forests (RF). We also aim to use the data collected from the SPHERE house, which is a multisensor platform, with various ML techniques to assess the reliability of an

automatic learning approach for modelling and analysing PD. As mentioned before, the data that will be used to learn from include the output of wearable accelerometer, RGB-D cameras with silhouette extraction, PIR and environmental sensors.

To achieve such analysis, multiple potential approaches could be considered for future work, some of which are presented in the following:

PD versus non-PD recognition—given the input data corresponding to different time scales, such as seconds, minutes or hours, ML-based model could be trained to discriminate between PD and non-PD (control) subjects. Such recognition could be performed as binary classification or as a probability estimation per category. Further, the predictions corresponding to different durations could be compared to determine the one by which the most reliable recognition is achieved.

PD severity estimation—based on different rating scales such as the MDS-UPDRS or H&Y score, a ML-based approach could learn to estimate the severity of the disease among the patients with PD. The ground-truth scores are provided by the clinicians during both 'on' and 'off' medication periods. Then, a model could be trained to regress predictions using the ground-truth scores.

Fine-grained analysis of PD symptoms—in addition to PD recognition and severity estimation, an ML-based model could be trained to detect different PD symptoms, such as bradykinesia or tremor. Also, a finer-grained analysis of each symptom could be investigated. For example, a model could learn to detect peaks or changes during tremor.

Activity recognition—the learning could be extended towards recognising the category of the activities which are performed by the subjects during free living or scripted activity experiments. Then, the quality of the performed activities, such as meal preparation or cleaning, could be

assessed. Also, the relationship between PD symptoms like bradykinesia and the performed activities could be modelled.

Behaviour analysis of the patients with PD—ML algorithms could be applied to model the behavioural patterns of subjects with PD versus subjects without PD. Such models could be also trained to distinguish between the behaviour of the patients with PD during 'on' and 'off' medication periods. Moreover, the patterns in the sleeping habits, mobility, and urinary function of the patients could be learnt and analysed.

Modelling the progression of PD temporally—the above-mentioned predictions could be monitored through time by modelling the progression of the disease via ML algorithms that provide temporal analysis of the input data. Specifically, models like HMMs or Recurrent Neural Networks, which can produce predictions per time point, could be trained to temporally model the PD symptoms. Therefore, for example, PD symptoms could be assessed at different times of the day during the study.

Analysing the information collected from screening and diaries—the questionnaires and tests performed during screening (contact B), as well as the diaries filled by the patients during their stay in the SPHERE house, would provide information which could be summarised via descriptive statistics. Additionally, associations with the data collected from various sensors, for example, environmental sensors, could be modelled via ML.

Evaluating traditional versus state-of-the-art ML algorithms—as presented in the article by Belić and colleagues,[34] there are many ML algorithms which can be applied to model PD. We would have the ability to assess the performance of such alternative algorithms. Specifically, a comparison between classical ML algorithms, such as SVMs or RFs, against state-of the-art deep learning-based approaches for modelling PD is of interest, as the latter has achieved high performance in areas such as computer vision.

Evaluating single versus multimodal approaches—due to the multisensor data collection platform within the SPHERE house, the performance of ML models which are based on single or multiple modalities could be assessed. In a multimodal approach, methods for fusing data from different modalities, for example, early or late fusion methods, could be also evaluated to determine which modalities are most effective.

### Qualitative data collection and analysis
A semistructured interview will be conducted with each participant at the end of the study exploring their experience of the SPHERE technology, with a focus on the tolerability and acceptability of such technology for people with PD and those close to them. An interview topic guide is shown in box 1. Interview transcripts will be read, reread and the data gathered will be analysed thematically, using an iterative–inductive approach,[35] with methods drawn from Braun and Clarke.[36]

---

**Box 1  Interview topic guide**

► Background experience with technology.
► Experience of the Sensor Platform for HEalthcare in a Residential Environment (SPHERE) sensor platform.
► Thoughts about the sensor data collected.
► The study information received relating to the SPHERE technology.
► Questions about the study experience specific to Parkinson's disease (PD) and those close to people with PD.

---

Additionally, two questionnaires will be used to augment the qualitative data produced from the interviews. The first questionnaire will be a standardised tool evaluating the participants' attitudes towards and experience of technology: using four attitude-based subscales (looking at positive attitudes, negative attitudes, technological anxiety/dependence and attitudes towards task-switching) from the Media and Technology Usage and Attitudes Scale.[37] This will allow us to interpret the interview responses in the context of a participant's prior background exposure to technology. The second questionnaire comprises three short questions exploring feedback about the sensors or study in general regarding thoughts that could have occurred to the participant in the days after they left the SPHERE house. These questions are listed in box 2. This enables us to ensure we understand the experience of this sensor platform as fully as possible.

### Data management and security
The day-to-day management of the study including data management will be coordinated by the principal investigator of the study. All the data collected will be fully anonymised or pseudonymised where possible. Unique study identification numbers will be assigned to each participant. Access to data will be limited to SPHERE researchers and collaborators only, who must be registered (to use the data) with managed access and permissions. Directly identifiable data (eg, name, address, telephone number) will be held securely and separately from other study data. Data procedures will be in accordance with the General Data Protection Regulation. If a participant withdraws, the data that are already obtained will be kept by the study team. Under a Service Level Agreement between the University of Bristol and University Hospitals Bristol NHS Foundation Trust, the Trust will monitor 10% of the university's sponsored studies. In line with the National Institute for Health and Research guidance which encourages sharing anonymised datasets, we will seek informed consent from participants for their anonymous data to be shared with other researchers. The data management will be reviewed on a regular basis by an in-house Data Monitoring Committee consisting of data experts, engineers and movement disorders clinicians.

---

**Box 2    Questions following up the SPHERE house stay (contact E questionnaire)**

► Following the PD SENSORS study, now that you have finished the study and been at home for a while, do you have any further thoughts to feedback to us about the SPHERE sensors that you lived with for 5 days?

► Specifically, in relation to your PD/supporting role for a person with PD, was there anything further you wanted us to know about or understand?

► Is there anything else at all that you felt important to mention to us regarding your study experience?

PD, Parkinson's disease; PD SENSORS, Parkinson's Disease Symptom Evaluation in a Naturalistic Setting producing Outcome measuRes using SPHERE technology; SPHERE, Sensor Platform for HEalthcare in a Residential Environment.

## Further study detail
### Collaboration with IXICO

The University of Bristol and IXICO will collaborate on specific outcomes, as well as through the sharing of data and results, the loan of the wearable biosensors (Axivity AX3 and GENEActiv Original) and gait mat (Zeno Walkway) and in-kind contributions of scientific expertise and study operations. The topics explored within this collaboration will include the feasibility of labelling total body bradykinesia from colour video, to developed automated detection of bradykinesia from biosensor data, biosensor device comparability, and relationships between sensor-based measurements from the gait mat and wearable devices and between sensor-based measurements and traditional clinical measures of motor symptoms, in addition to ADL detection from wearable sensor data.

## Safety reporting

Any adverse event (AE) will be assessed and acted on by the research team. All AEs will be recorded in the study or project file with a note that will identify when the event occurred, the details of the AE, any potential study relation, action taken and resolution/ closure of the AE. An assessment of seriousness will be made by the researcher and serious AEs (SAEs) will be reported in line with current legislation and university guidance.

In addition, all SAEs will be reported to the NHS Research Ethics Committee in the Annual Progress Report.

## Amendments

Any amendments to protocol design will be communicated to relevant parties including North Bristol NHS Trust and the Research Ethics Committee.

## Patient and public involvement

The Bristol Movement Disorders Health Integration Team (MOVE-HIT) has an active PPI group which includes people with PD and their partners/carers. The MOVE-HIT PPI group meets regularly and has

been involved in providing extensive advice around the study design, as well as the development of the participant information sheets, consent forms and participant diaries. The opinions of members of this PPI group on study design have been collected both in face-to-face group and individual consultations and in a survey to wider group members. We plan further ongoing consultation, both during the middle of the study to discuss the group's views on study progress and at the end of the study to consult on the plain English report on the study findings and to identify dissemination outlets.

The PD SENSORS research team are also collaborating with Lyndsey Isaacs, a Trustee of the CPT, who facilitated the presentation of a PD SENSORS study design survey to the delegates at a CPT Research Update Meeting. This survey was also made more widely available on the CPT website. The research team will have ongoing liaison with CPT representatives during the study to discuss aspects of study management such as dissemination of research findings. The CPT have agreed to be a forum through which to disseminate the study poster to allow interested persons to express interest in participating in PD SENSORS.

## Dissemination

It is anticipated that the scientific results of the study and its various elements will be published both via presentation at national or international conferences and through publication in peer-reviewed scientific journals/publications.

**Author affiliations**
[1]Translational Health Sciences, University of Bristol Medical School, Bristol, UK
[2]Movement Disorders Group, North Bristol NHS Trust, Avon, UK
[3]School of Computer Science, Electrical and Electronic Engineering and Engineering Mathematics, Faculty of Engineering, University of Bristol, Bristol, UK
[4]Research and Development, IXICO, London, UK
[5]Population Health Sciences, University of Bristol Medical School, Bristol, UK
[6]Institute of Neuroscience, Newcastle University, Newcastle, UK
[7]NHS Foundation Trust, Newcastle Upon Tyne Hospitals, Newcastle Upon Tyne, UK
[8]Department of Neurology, University Medical Center Schleswig-Holstein Campus Kiel, Kiel, Germany
[9]Research Department, Cure Parkinson's Trust, London, UK
[10]Project Management, Bristol Health Partners, Bristol, UK

**Contributors**  CM drafted the study protocol, worked with all contributors to improve and finalise details and continues as the study's principal investigator. IC and AW supervise the studies and work of CM; AW is the study's chief investigator, IC is the principal of SPHERE-IRC; both have major influence on the study's direction. ELT has advised on data management. KK and RoM have done pilot work and have advised on study direction. MM, FH, RMC, ELT and SW have been involved in the computer science and analytics planning. JC, AH, MR and RE have given practical input on study tools and documentation. OW and HM are representatives of the Public and Patient Involvement groups which have had strong influence on study design. LR and WM have added expertise in the field to advise CM on early study design.

**Funding**  This work was performed under the SPHERE Next Steps Project funded by the UK Engineering and Physical Sciences Research Council (EPSRC), Grant EP/R005273/1. This work is supported by the Elizabeth Blackwell Institute for Health Research, University of Bristol and the Wellcome Trust Institutional Strategic Support Fund, grant code: 204813/Z/16/Z, by The Cure Parkinson's Trust, grant code AW021 and by IXICO, grant code R101507-101. Dr Jonathan de Pass and Mrs Georgina de Pass made a charitable donation to the University of Bristol (of which Jonathan is a

---

graduate) through the Development and Alumni Relations Office to support research into Parkinson's Disease. The funding pays for the salary of CM, the PhD student and Clinical Research Fellow, and she reports to the donors on her progress.

**Conflicts of interests** KK is employed by IXICO, that have provided some financial support for the study. Their search may lead to the development of products which may be licensed to IXICO.

**Patient consent for publication** Not required.

**Provenance and peer review** Not commissioned; externally peer reviewed.

**ORCID iDs**
Catherine Morgan http://orcid.org/0000-0003-0333-2417
Michal Rolinski http://orcid.org/0000-0003-1191-7060

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
