## [Reviewer comments · BMJ Open]

ARTICLE DETAILS

TITLE (PROVISIONAL)	Protocol for PD SENSORS: Parkinson's Disease Symptom Evaluation in a Naturalistic Setting producing Outcomes measuRes using SPHERE technology. An observational feasibility study of multi-modal multi-sensor technology to measure symptoms and activities of daily living in Parkinson's disease.
AUTHORS	Morgan, Catherine; Craddock, Ian; Tonkin, Emma; Kinnunen, Kirsi; McNaney, Roisin; Whitehouse, Sam; Mirmehdi, Majid; Heidarivincheh, Farnoosh; McConville, Ryan; Carey, Julia; Horne, Alison; Rolinski, Michal; Rochester, Lynn; Maetzler, Walter; Matthews, Helen; Watson, Oliver; Eardley, Rachel; Whone, Alan

VERSION 1 – REVIEW

REVIEWER	Salama A. Mostafa Universiti Tun Hussein Onn Malaysia
REVIEW RETURNED	29-Jul-2020

GENERAL COMMENTS	>My comments to the protocol that is titled "Protocol for PD SENSORS: Parkinson's Disease Symptom Evaluation in a Naturalistic Setting producing Outcomes measures using SPHERE technology. An observational feasibility study of multi-modal multi-sensor technology to measure symptoms and activities of daily living in Parkinson's disease" are as follows: >>The second part of the abstract needs improvement to include the processing steps of the methodology and the finding. >>The points of strengths and limitations should be divided to two parts or put in a table of two columns so they can be easily identified. >>in limitation number two, it is mentioned that the work cover mild to moderate PD so it will not be generalized to people with more severe disease. Is it because of the selected people involved in the testing? usually when the system is able to detect mild to moderate PD from their daily activity patterns should be able to identify severe cases. >>The authors view the proposed solution as an alternative solution to the clinical rating scales approach but in my opinion they should complement each others. >>The methods are well presented. However, there is no clear explanation to the research methodology and the related processes. They will be useful for other researchers to follow. >>The design only explained by one sentence. The authors might elaborate on the protocol design and assessment flow. >>The way that the protocol is written is too specific for the use in SPHERE house which is acceptable. However, some parts can be
---

	generalized to fit with other applications as the SPHERE house modeling is common in IoT based environments.
--	--

REVIEWER	Amin Cervantes-Arriaga Instituto Nacional de Neurología y Neurocirugía
REVIEW RETURNED	10-Aug-2020

GENERAL COMMENTS	The abstract has a good introduction but other than that is not useful. After reading the abstract I had no idea about the study design, study sample, or selected outcomes measures. The abstract should be rewritten to be more informative even if the paper is a protocol description. The introduction is well written highlighting the background, problem and rationale. The SPHERE platform is described and appropriate citations included. Study protocol is defined as feasibility and acceptability study. The main objective is understanding how the raw data can be processed into meaningful data from the clinical standpoint. While the understanding is kind of unspecific it still is clear enough given the context, Secondary objectives are correct. Regarding the methods, a more detailed description of the "SPHERE house" would help identifying how and which activities of daily living could be assessed. Inclusion/exclusion criteria are OK. The study procedures including the semi-structured interview are well described. Data management is included, as well as adverse events protocols. Suggestions: Rewrite the abstract. Better description of the SPHERE house. Perhaps add informed consent in the inclusion criteria. Adding a brief description of all the clinical tools (BDI, NMSS, PDQ-39, RBD-SQ, etc).
---

VERSION 1 – AUTHOR RESPONSE

Reviewer(s)' Comments to Author:

Reviewer: 1

Reviewer Name: Salama A. Mostafa

Institution and Country: Universiti Tun Hussein Onn Malaysia

Competing interests: None.

My comments to the protocol that is titled "Protocol for PD SENSORS: Parkinson's Disease Symptom Evaluation in a Naturalistic Setting producing Outcomes measures using SPHERE technology.

An observational feasibility study of multi-modal multi-sensor

technology to measure symptoms and activities of daily living in Parkinson's disease" are as follows:

Reviewer 1 comment 1: The second part of the abstract needs improvement to include the processing steps of the methodology and the finding.

Author response: We agree that this would help readers of the abstract by enhancing the overview of the study and its methodology. We have amended the second part of the abstract, which now reads:

“Methods and Analysis: This is a small-scale feasibility and acceptability study during which 12 pairs of participants (comprising a person with Parkinson’s and a healthy control participant) will stay and live freely for 5 days in a home-like environment embedded with SPHERE technology including environmental, appliance monitoring, wrist-worn accelerometry and silhouette video sensors. These data will be collected alongside clinical rating scales, participant diary entries and expert clinician annotations of colour video images. Machine learning will be used to look for a signal to discriminate between Parkinson’s disease and control, and between Parkinson’s disease symptoms ‘on’ or ‘off’ medications. Additional outcome measures including bradykinesia, activity level, sleep parameters and some activities of daily living will be explored. Acceptability of the technology will be evaluated qualitatively using semi-structured interviews.”

Reviewer 1 comment 2: The points of strengths and limitations should be divided to two parts or put in a table of two columns so they can be easily identified.

Author response: This is a good idea and, within the guidance from the Editor of BMJ Open (“This section should contain five short bullet points, no longer than one sentence each, that relate specifically to the methods.”), we have now divided the ‘Strengths and limitations of this study’ section into separate sections for strengths and limitations.

Reviewer 1 comment 3: in limitation number two, it is mentioned that the work cover mild to moderate PD so it will not be generalized to people with more severe disease. Is it because of the selected people involved in the testing? usually when the system is able to detect mild to moderate PD from their daily activity patterns should be able to identify severe cases.

Author response: We agree that it is possible that the machine learning algorithms may be able to be extrapolated to identify severe cases, although this will not be validated in this study protocol, so we have removed that limitation from the protocol.

Reviewer 1 comment 4: The authors view the proposed solution as an alternative solution to the clinical rating scales approach but in my opinion they should complement each others.

Author response: We agree wholeheartedly that direct clinician-patient interaction is irreplaceable and vital in the care of people with Parkinson’s disease and in clinical trials in movement disorders. To clarify our thoughts on this, we have added the following sentence to the final paragraph of the Introduction section:

“SPHERE technology has the potential to complement and enhance the existing clinical rating scales used to evaluate Parkinson’s disease.”

Reviewer 1 comment 5: The methods are well presented. However, there is no clear explanation to the research methodology and the related processes. They will be useful for other researchers to follow.

Author response: Thank you for saying that the methods are well-presented. In response to your comment that you felt there was not a very clear explanation to the research methodology, we have taken a number of actions:

We have moved the 'Statistical analysis' section which details the computer science approaches and methodology in data analysis to earlier in the paper, hopefully providing a better sense of continuity from the description of how the study is going to be carried out.

We agree that the 'Schedule of Contact D: the stay in the SPHERE house' section is not very clear and therefore not easy for others to reproduce in the future. Therefore, we have reworked this section somewhat and added the following sentences:

"During the time they are in the house, the participants will be encouraged to continue living freely and undertake their regular activities of daily living as normally as possible."

"Researchers will visit the house on two occasions, Day 1 and Day 4, to conduct scripted activities and clinical rating scales with participants"

"During Days 2 and 3 the participants will live freely with no researcher visits."

"Specific food preparation and cleaning/hygiene scripted activities will be witnessed and annotated by a study team member during their planned visits to the house. The participant with PD will undertake these activities on Day 1 and twice on Day 4, first 'off' then 'on' medications, whilst the control participant will do the activities once each on Day 1 and Day 4." More detail about these activities is shown in the supplementary material.

"Wearable devices: in addition to the SPHERE wearable device on each wrist, an extra IXICO-provided wearable will be worn on each wrist throughout the SPHERE house stay. During the clinical assessments and scripted activities on day 4 one additional wearable accelerometer will be placed around the trunk to locate in the lower back (lumbar) area and another one on a lower limb (shin)."

"Short periods of these [colour video] data, collected both while participants are free-living and during clinical assessments, will be used to provide a labelled dataset for the analytic work-up of total body movement using annotations of this dataset relating to the participants' activities, location, and aspects of their Parkinson's disease symptomatology."

We hope that these measures, in addition to Figure 2 which gives a visual overview of Contact D, this has made our methodology clearer to readers of the article.

Reviewer 1 comment 6: The design only explained by one sentence. The authors might elaborate on the protocol design and assessment flow.

Author response: We hope that the steps taken above have gone some way towards aiding understanding of the assessment flow and protocol design. Figure 1 is designed to help a reader understand broadly what is taking place at each study 'Contact', and Figure 2 gives an overview of Contact D.

Reviewer 1 comment 7: The way that the protocol is written is too specific for the use in SPHERE house which is acceptable. However, some parts can be generalized to fit with other applications as the SPHERE house modeling is common in IoT based environments.

Author response: The reworking of the description of Contact D should hopefully allow our protocol to be applied to other multi-modal, multi-sensor environments by other groups.

Reviewer: 2

Reviewer Name: Amin Cervantes-Arriaga

Institution and Country: Instituto Nacional de Neurología y Neurocirugía

Competing interests: None declared

Please leave your comments for the authors below

The abstract has a good introduction but other than that is not useful. After reading the abstract I had no idea about the study design, study sample, or selected outcome measures. The abstract should be rewritten to be more informative even if the paper is a protocol description.

The introduction is well written highlighting the background, problem and rationale.

The SPHERE platform is described and appropriate citations included.

Study protocol is defined as feasibility and acceptability study.

The main objective is understanding how the raw data can be processed into meaningful data from the clinical standpoint. While understand is kind of unspecific it still is clear enough given the context, Secondary objectives are correct.

Regarding the methods, a more detailed description of the "SPHERE house" would help identifying how and which activities of daily living could be assessed.

Inclusion/exclusion criteria are OK. The study procedures including the semi-structured interview are well described. Data management is included, as well as adverse events protocols.

Author response: Thank you for your kind words and for taking the time to read our protocol in detail and provide really helpful suggestions.

Suggestions:

Reviewer 2 comment 1: Rewrite the abstract.

Author response: We agree that the abstract needed strengthening with more information for the reader. We have therefore re-written the Methods and Analysis section, so that it now reads:

“Methods and Analysis: This is a small-scale feasibility and acceptability study during which 12 pairs of participants (comprising a person with Parkinson’s and a healthy control participant) will stay and live freely for 5 days in a home-like environment embedded with SPHERE technology including environmental, appliance monitoring, wrist-worn accelerometry and silhouette video sensors. These data will be collected alongside clinical rating scales, participant diary entries and expert clinician annotations of colour video images. Machine learning will be used to look for a signal to discriminate between Parkinson’s disease and control, and between Parkinson’s disease symptoms ‘on’ or ‘off’ medications. Additional outcome measures including bradykinesia, activity level, sleep parameters and some activities of daily living will be explored. Acceptability of the technology will be evaluated qualitatively using semi-structured interviews.”

Reviewer 2 comment 2: Better description of the SPHERE house.

Author response: Thanks, we agree it would be interesting to the reader to understand the testing location better. We have included the sentence below within the Introduction ‘Sensor Platform for Healthcare in a Residential Environment – SPHERE’ section:

“The SPHERE sensor platform is embedded into a test-bed location called the ‘SPHERE house’. This is a 2-bedroom, 2-storey property with kitchen, living room, dining room, bathroom and toilet.”

Reviewer 2 comment 3: Perhaps add informed consent in the inclusion criteria.

Author response: Thank you, we added a sentence in the ‘Setting and Participants’ section to state that informed consent was a pre-requisite to participating in the study:

“The study will be conducted in accordance with the Declaration of Helsinki and Good Clinical Practice Guidelines. Written informed consent will be obtained from all study participants before any study-related procedures are performed.”

Reviewer 2 comment 4: Adding a brief description of all the clinical tools (BDI, NMSS, PDQ-39, RBD-SQ, etc).

Author response: This is a great idea, thanks. We have added the following text about the clinical tools and questionnaires used in the study to the Methods section:

“Several clinical rating scales will be completed as described at Contact B. This is with the intent to gain a thorough understanding of the nature of the symptoms experienced by the participants with PD, and to be able to therefore make better sense of the sensor data produced from the SPHERE house stay. The Parkinson’s Disease Sleep Scale-2 is a 15-point visual analogue scale that has been validated in PD (29) which allows patients with PD to self-rate and quantify the profile of nocturnal disturbances and sleep disruption being experienced. The Non-Motor Symptoms Scale (NMSS) is a 30-question scale which assesses severity and frequency with which a patient with PD experiences a variety of non-movement related symptoms, such as urinary dysfunction, mood/cognitive symptoms and gastrointestinal tract-associated symptoms (30). The Timed Up and Go (TUG) test (31) is a simple, quick and widely-used clinical performance tool which assesses lower extremity function, mobility and falls risk. The REM sleep Behaviour Disorder Screening Questionnaire (RBD-SQ) is a useful screening tool due to its relatively high sensitivity for REM sleep behaviour disorder (32), which affects a significant proportion of patients with PD and may change the interpretation of the sensor data overnight.

During Contact D, the research team will conduct the Movement Disorder Society sponsored revision of the Unified Parkinson’s Disease Rating Scale (MDS-UPDRS) (5) with the participant with Parkinson’s. This clinical rating scale has subscales evaluating motor experiences of daily living (part I), non-motor experiences of daily living (part II), motor examination (part III) and motor complications (part IV). The PDQ-39, also to be used in Contact D, is a self-completed patient-reported outcome measure designed to address aspects of functioning and well-being for those affected by Parkinson’s disease (33).”

Author final comments:

We are very grateful to the Editor and both reviewers for your detailed review of this manuscript and helpful comments. We wish you all the best in these strange times.

VERSION 2 – REVIEW

REVIEWER	Salama A. Mostafa Universiti Tun Hussein Onn Malaysia
REVIEW RETURNED	10-Oct-2020

GENERAL COMMENTS	>>An improvement to the structure and layout of the reported protocol is needed. >>A clarification to the methods used in the qualitative and quantitative data collection and analysis is required. >>Some references need to be updated.
--

REVIEWER	Amin Cervantes-Arriaga Instituto Nacional de Neurología y Neurocirugía, Mexico
-----------------	---

REVIEW RETURNED	02-Oct-2020
GENERAL COMMENTS	All my concerns have been properly addressed. I also consider the reponse to the editor queries and the other reviewer to be adequate. I have no further suggestions.

VERSION 2 – AUTHOR RESPONSE

Reviewer 1

Comment 1: An improvement to the structure and layout of the reported protocol is needed.

Author response:

We have re-looked at the section order of this manuscript in light of your comment recommending improvement to the structure and layout of the protocol, and have made the following changes:

1. Moved the 'Collaboration with IXICO plc' section to below the 'Qualitative data collection and analysis' section in order to keep all the data analysis methodology together in one section.
2. Moved the 'Sample size' section to just below the 'Setting and participants' section in order to answer any immediate reader interest as to why that sample size was chosen.
3. Inserted an over-arching title named 'Data Analysis and Management' which encompasses the Statistical (quantitative) and qualitative data analysis sections. Also this includes the 'Data management and security' and 'Data statement' paragraphs which are now placed together, to keep all data-related sections contiguous.
4. A final overarching title 'Further study detail' includes the sections 'Collaboration with IXICO plc', 'Safety reporting', 'Amendments', 'Patient and public involvement', 'Ethics and dissemination' and 'Funding'.

→ The manuscript's main sections now read like this: Abstract – Introduction – Aims – Methods – Data Analysis and Management – Further Study Detail. We hope that these improvements make the paper easier to navigate and more user-friendly.

Comment 2: A clarification to the methods used in the qualitative and quantitative data collection and analysis is required.

Author response:

To answer your comment as fully as possible, we have divided it into responses relating to data collection (quantitative and qualitative) and data analysis (quantitative and qualitative).

Data collection:

Quantitative

1. We have recorded what the electronic sensors which record data are within the Introduction section under 'Sensor Platform for HEalthcare in a Residential Environment – SPHERE'

"Within the SPHERE house there are a number of sensors which include: wearable devices with accelerometers (in-house developed); environmental/ambient sensors which measure temperature, humidity and light levels; RGB-D ('red green blue – depth', referring to the three colour planes that together can generate the colour of a pixel in an image, in addition to the recording of distances to

surfaces using infra-red light) camera sensors that generate silhouettes of participants and bounding boxes that, when linked with the SPHERE wearable, let the system know who the participant is; appliance sensors which are able to tell when and how frequently an appliance is switched on and off; PIR (Passive InfraRed) sensors which detect motion in each room.”

2. To clarify which sensors are recording data at what stage during the study, we have added the following text highlighted in yellow to the Methods section under ‘Schedule of Contact D: the stay in the SPHERE house’:

- “Continuous sensor data will be captured throughout the 5-day stay from the following SPHERE platform sensors: silhouette and bounding box cameras, appliance sensors, environment/ambient sensors, PIR sensors.
- Wearable devices: in addition to the SPHERE wearable device on each wrist recording continuous accelerometry, an extra IXICO-provided wearable will be worn on each wrist throughout the 5-day stay. During the clinical assessments and scripted activities on day 4 one additional wearable accelerometer will be placed around the trunk to locate in the lower back (lumbar) area and another one on a lower limb (shin).”

3. We have also given more detail about the gait tests which will be undertaken on the gait mat (highlighted in yellow):

“During the ‘off’ and ‘on’ medication testing session on Day 4, the participants will undertake clinical assessments including an evaluation of gait (normal pace, slow pace and fast pace) on the Zeno Walkway, a gait mat embedded with sensors that is placed in the entrance area of SPHERE house where these assessments are conducted.”

4. With regards to the clinical rating scales, we have added a sub-section title “Clinical Rating Scales” to better signpost the reader towards which scales we are deploying in the study and why. We believe that this section, alongside Table 1 (“Baseline and follow-up data from Contacts B and D”) and the “Schedule of Contact D: the stay in the SPHERE house”, will give the reader a good overview of how we are undertaking quantitative data collection. However, as is the nature of exploratory feasibility work, this is an iterative process and therefore giving more information than this may lead to inaccuracies/mislead the reader.

Qualitative

1. To better signpost the reader towards what the ‘Qualitative section’ includes, we have renamed it “Qualitative data collection and analysis”.
2. There is detail about the nature of the interview, the interview topic guide and the specific questionnaires to be used in the qualitative section (including the reference of the first questionnaire and the script of the second questionnaire in Box 2).

Data analysis:

Quantitative

We agree that, for other researchers looking to learn from or reproduce our work, the more detail the better with regards to the quantitative/computer science analytic approach to the data. However, due to the nature of this study being an iterative and exploratory pilot and feasibility study, we are not in a position to detail exactly which analytic techniques will be deployed to produce specific outcomes.

We hope that when we have had a chance to deploy our machine learning approaches on the data (when it has been collected), we will be able to describe to other researcher much more explicitly how our outcomes had been achieved

To make this point clear to the reader, we have added the following (highlighted) text at the beginning of the (renamed) “Statistical data analysis approaches” section:

“The purpose of this preliminary and exploratory study is to explore the best way to quantify PD using the in-house technology, therefore the exact nature of the analytic approach(es) is not available at this stage. Study researchers and the study statistician will work collaboratively with the clinicians and SPHERE machine learning team to determine how the SPHERE data can be transformed into usable form...

We go on to give multiple examples of potential machine learning approaches to analysing the data.

Qualitative

At this stage, before reading any interview transcripts, we are still intending to use the inductive approach to qualitative analysis with reference to methods by Braun and Clarke, as stated in the manuscript. However, we are likely to also draw from methods outlined by O’Reilly including the iterative-inductive approach to evaluating sets of transcripts. Therefore, in addition to including our interview topic guide, notes on how the two study questionnaires will augment and enhance our understanding of acceptability which we are seeking through the interviews, and the notes already made on the qualitative approach, we have included the following extra information:

“Interview transcripts will be read, re-read, and the data gathered will be analysed thematically, using an iterative-inductive approach (35), with methods drawn from Braun and Clarke (36).”

35. O’Reilly K. *Ethnographic methods* [second edition]. *Ethnographic Methods*. London: Routledge; 2012.

Comment 3: Some references need to be updated.

Author response:

1. The paper by Artusi et al which had been referenced as 2018 in *Movement Disorders* as a supplement has now been updated to the 2020 online publication by the same authors:

“Artusi, C.A., Imbalzano, G., Sturchio, A. et al. Implementation of Mobile Health Technologies in Clinical Trials of Movement Disorders: Underutilized Potential. *Neurotherapeutics* (2020). <https://doi.org/10.1007/s13311-020-00901-x>”

2. The systematic review about technology originally quoted has been replaced by the more recent review about technology in the free-living environment published in 2020 by the authors of this manuscript:

“Morgan C, Rolinski M, McNaney R, et al. Systematic Review Looking at the Use of Technology to Measure Free-Living Symptom and Activity Outcomes in Parkinson’s Disease in the Home or Home-like environment. *Journal of Parkinson’s Disease* 2020;10:429-454.”